# Joint Representation Training in Sequential Tasks with Shared Structure

## Abstract

Classical theory in reinforcement learning (RL) predominantly focuses on the single task setting, where an agent learns to solve a task through trial-and-error experience, given access to data only from that task. However, many recent empirical works have demonstrated the significant practical benefits of leveraging a joint representation trained across multiple, related tasks. In this work we theoretically analyze such a setting, formalizing the concept of *task relatedness* as a shared state-action representation that admits linear dynamics in all the tasks. We introduce the Shared-MatrixRL algorithm for the setting of Multitask MatrixRL Yang & Wang (2020). In the presence of $P$ episodic tasks of dimension $d$ sharing a joint $r \ll d$ low-dimensional representation, we show the regret on the the $P$ tasks can be improved from $O(PHd\sqrt{NH})$ to $O((Hd\sqrt{rP} + HP\sqrt{rd})\sqrt{NH})$ over $N$ episodes of horizon $H$. These gains coincide with those observed in other linear models in contextual bandits and RL Yang et al. (2020); Hu et al. (2021). In contrast with previous work that have studied multi task RL in other function approximation models, we show that in the presence of bilinear optimization oracle and finite state action spaces there exists a computationally efficient algorithm for multitask MatrixRL via a reduction to quadratic programming. We also develop a simple technique to shave off a $\sqrt{H}$ factor from the regret upper bounds of some episodic linear problems.

## 1 Introduction

Reinforcement learning (RL) is about learning via doing – learning to solve a sequential decision-making task where the only information about the task is obtained via trial-and-error. Accordingly, the underlying assumptions made in RL are typically minimal. Beyond what can be learned from trial-and-error experience, the learner's structural prior on the underlying task is commonly restricted to a small set of Markov assumptions (Puterman, 1990): namely, that the task is of a sequential nature, with the task reward and state transition dynamics at each step determined by an (unknown) Markov process.

The simplicity of this setting, which forms the basis of a rich and diverse literature (Bertsekas, 2019; Sutton, 1992), stands in contrast to the complexity of many real-world settings, where one has access to data from multiple, related tasks. In these situations, experience from one task can often be leveraged to accelerate learning in another. For example, when humans are confronted with learning a new video game, we naturally draw on previous experience and knowledge from playing other games, even if the dynamics and rewards across the games are not the same.

In line with this intuition, there exist a number of empirical works which demonstrate how experience can be gathered from multiple tasks to accelerate RL over learning these tasks in isolation. For example in robotics (Yu et al., 2020; Kalashnikov et al., 2021), such approaches are key to avoiding an expensive blow-up in the sample complexity of required, real-world interactions. A popular paradigm to jointly use experience from multiple tasks is by way of learning a *shared low-dimensional representation*. Namely, the observations of each task are individually embedded into a common low-dimensional space, and learning occurs jointly in this space. (Teh et al., 2017; D'Eramo et al., 2019).

Despite these empirical successes, theoretical explorations to understand the benefits of such *joint training* in RL have been limited. While in the supervised learning literature, the benefit of multi-task training is well-studied (e.g., Baxter, 1995; 2000; Ben-David & Schuller, 2003; Du et al., 2020; Tripuraneni et al., 2020; 2021), obtaining a similar understanding in the setting of RL is more challenging. For one, a sufficiently flexible yet useful notion of "task relatedness" is difficult to formulate in RL, which involves both rewards and transition dynamics. Secondly, an algorithm using such a relatedness measure must carefully balance exploration and exploitation, while appropriately handling inevitable inaccuracies in the learned representation and how these can compound over the horizon.

Given these existing shortcomings in the literature, in this work we aim to theoretically analyze the benefit of learning joint representations for multi-task RL. We begin by formalizing the underlying similarity – i.e., *task relatedness* – between multiple tasks. Leveraging recent results on *linearly factored* or *low-rank* MDPs (Agarwal et al., 2020; Yang & Wang, 2020; Nachum & Yang, 2021), we assume that there exists a state-action representation such that all tasks admit linear transition dynamics with respect to this representation. In this setting, any one task may exhibit distinctly different dynamics from the remaining tasks while still maintaining a common and learnable structure. Under such a shared representation, we quantify the benefit – in terms of regret – given by using a sufficiently accurate approximate representation, and we pair this result with an online algorithm for simultaneously learning and using such a representation. Our results provide a clear understanding of the trade-offs associated with leveraging a jointly learned representation in the setting of RL as a function of the dimension of the shared representation, the number of tasks, and the dimension of the raw state observations (see our main results in Section 4). We show that in the presence of bilinear optimization oracle there exists a computationally efficient algorithm for multitask MatrixRL via a reduction to quadratic programming (see Section 5). This is in contrast with previous work that have studied multi task RL in other function approximation models such as Hu et al. (2021). We develop a general regret analysis technique to shave off a $\sqrt{H}$ factor from the the regret upper bounds of episodic linear problems and apply it both to the original MatrixRL rates (see Section 3) as well as Shared-MatrixRL (see Section 4).

## 2 Related Work

As mentioned above, multi-task learning is well-studied in the supervised learning literature. The predominant mechanism for performing multi-task learning in these previous works is analogous to our own, namely, parameterizing a classifier as a composition of two functions, one of which is common to all tasks and another which is unique to each class (Baxter, 1995; Maurer, 2006; Du et al., 2020; Tripuraneni et al., 2020; 2021). As in our own work, these previous works generally rely on an assumption that an optimal hypothesis with the desired compositional form exists, although some work has explored alternative assumptions (Ben-David & Schuller, 2003).

More closely related to our own setting is the work of D'Eramo et al. (2019), which theoretically analyze approximate dynamic programming in the context of multi-task learning. In this setting, an approximate value function is learned, with a common representation used to parameterize this value function. However, it is important to note that approximate dynamic programming is distinct from RL, as it ignores the difficult exploration problem associated with learning from one's own collected data. In contrast, our analysis is specifically tailored to an online learning scenario: where one of the main challenges is deriving multi-task learning bounds which carefully balance exploration and exploitation jointly across all tasks.

Works that do consider the online learning setting include Yang et al. (2020) and Hu et al. (2021). The first of these considers a linear bandit setting and is not immediately applicable to RL; moreover, this work imposes additional structural conditions on the linear features of the bandit problem which effectively require the action features to sufficiently cover all possible directions. The second work by Hu et al. (2021) is closer to ours and considers the linear RL setting. Our 'sharedness' assumptions and results are a generalization of those studied in Hu et al. (2021). Because Hu et al. (2021) studies a value-based approach, the assumption is that all the underlying linear parameter weights of the task's $Q$-functions lie in (or near) the same subspace. Because we are studying a model based approach we move beyond this sharedness structure to instead study the setting where the task model matrices share a common factorization. This is the reason our bounds have a dependence on $d'$ as well as on $d$ and $r$. Our regret guarantees are similar to those in Hu et al.

(2021), despite taking a different approach with distinct derivations. We don't see this as a limitation but rather as an indication to the wider research community that there is a potential opportunity to develop a unifying analysis of RL methods in the presence of shared representation learning that could subsume both value-based and model-based methods. We believe this to be an interesting and exciting avenue for future research. Moreover, we show (see Section 5) that as long as we have access to an oracle for joint least squares matrix factorization, the optimization problem required to find the policy to execute at time $t$ can be solved efficiently. This is in stark contrast with other approaches where even solving the necessary joint optimization problem over the task family to find the policy to execute at any given time can be an intractable problem. Recently, other works Cheng et al. (2022); Agarwal et al. (2022); Lu et al. (2022) have explored more general versions of our shared task assumptions where the learner may have access only to a family of representation functions and is tasked with learning a viable representation while interacting with multiple tasks at once. They show that shared representation learning is advantageous when compared to learning a single representation per task. These are very close in spirit to the study we present here and should be thought of as a successor works to ours.

In this work we consider a model for task relatedness inspired by Tripuraneni et al. (2021), where we assume the underlying model of the MDP dynamics have a shared low rank representation. Other models of the relationship between related tasks are possible. Most notably Müller & Pacchiano (2022) and Moskovitz et al. (2022): in Müller & Pacchiano (2022) the authors consider the question of learning an appropriate 'bias' vector for regularizing the MatrixRL algorithm. This allows them to show that in case the variance of the models in the family is small, performance (in this case measured in the form of regret) in a test task can be substantially better. The authors of Moskovitz et al. (2022) tackle a similar issue. In their work they show that under the assumption that the optimal policies are similar across tasks in the family, it is possible to learn a useful default policy such that a policy gradient algorithm that regularizes towards it can learn an optimal policy for a target task much more efficiently than an algorithm regularizing towards the uniform policy. We leave the task of generalizing our work to the setting of a set or distribution train tasks for the purpose of solving a test task for future work.

## 3 Preliminaries

Formally, we consider the setting of episodic reinforcement learning proposed in (Yang & Wang, 2020) where an an agent explores an MDP $(\mathcal{S}, \mathcal{A}, \mathbb{P}, r, H)$ with state space $\mathcal{S}$, action space $\mathcal{A}$ and known reward function $r : \mathcal{S} \times \mathcal{A} \to [0, 1]$ whose transition dynamics are given by the feature embedding,

$$\mathbb{P}(\tilde{s}|s, a) = \phi(s, a)^\top \mathbf{M}_\star \psi(\tilde{s})$$

The learner receives a noiseless reward $r(s, a)$ which for simplicity we assume is known. All interactions between a policy and the MDP is of length $H$. For any policy $\pi$, state $s$, action $a$ and $h \in [H]$ we define $V_h^\pi(s), Q_h^\pi(s, a)$ as the value and $Q$ functions of policy $\pi$. Our objective is to design algorithms with small regret, defined as

$$R(NH) = \sum_{n=1}^{N} V_1^{\pi_\star}(s_{n,1}) - V_1^{\pi_n}(s_{n,1}),$$

Where $\pi_\star$ corresponds to the optimal policy, $\pi_n$ is the algorithm's policy during time-step $n$ and $s_{n,1}$ are the initial states during the $n$th episode.

The algorithm in Yang & Wang (2020) works by building an estimator $\widetilde{\mathbf{M}}_n$ of the matrix $\mathbf{M}_\star$ at time $n$ using the data collected so far. We use the notation $t = (n, h)$ (i.e. episode $n \leq N$ and stage $h \leq H$), to denote the state-action-state triplets $(s_t, a_t, \tilde{s}_t)$ where $\tilde{s}_t = s_{t+1}$. For simplicity we denote the associated features by:

$$\phi_t = \phi(s_t, a_t) \in \mathbb{R}^d, \ \psi_t = \psi(\tilde{s}_t) \in \mathbb{R}^{d'} \text{ and } \mathbf{M}_\star \in \mathbb{R}^{d \times d'}.$$

Denote $\mathbf{\Psi} \in \mathbb{R}^{|\mathcal{S}| \times d'}$ as the matrix whose rows equal $\psi(s)$ for all $s \in \mathcal{S}$ and let $\mathbf{K}_\psi = \sum_{\tilde{s}} \psi(\tilde{s}) \psi(\tilde{s})^\top$. For any matrix $\mathbf{B}$ we use $\mathbf{B}[:, i]$ to refer to $\mathbf{B}$'s $i-$th column. We use the notation $\| \cdot \|_F$ to denote the Frobenius norm of a matrix and $\| \cdot \|_2, \| \cdot \|_\infty$ the $l_2$ and $l_\infty$ norms of a vector. We will make the following assumptions regarding the norms of $\mathbf{M}_\star$ and the feature maps $\phi$ and $\psi$.

**Assumption 3.1** (Boundedness). *The feature maps $\phi$ and $\psi$ satisfy $\|\phi(s,a)\|_2 \leqslant L_\phi$, $\|\psi(s)\|_2 \leqslant L_\psi$ and $\|\mathbf{M}_\star[:,i]\|_2 \leqslant S$ for all $s, a \in \mathcal{S} \times \mathcal{A}$ and $i \in [d']$ and some known values $L_\phi, L_\psi$ and $S$. And therefore $\|\mathbf{M}_\star\|_F \leqslant \sqrt{d'}S$*

We also consider the following two assumptions on feature regularity, both present in Yang & Wang (2020).

**Assumption 3.2.** *[Feature Regularity] For all $\mathbf{v} \in \mathbb{R}^{|\mathcal{S}|}$, $\|\Psi^\top \mathbf{v}\|_\infty \leqslant C_\psi \|\mathbf{v}\|_\infty$, and $\|\Psi \mathbf{K}_\psi^{-1}\|_{2,\infty} \leqslant C'_\psi$, where $\|\mathbf{Y}\|_{2,\infty} = \max_i \sqrt{\sum_j \mathbf{Y}_{i,j}^2}$ is the $2, \infty$ norm (infinity norm over the $l_2$ norm of $\mathbf{Y}$'s columns).*

We will also prove sharper results under a more refined feature regularity assumption,

**Assumption 3.3.** *[Stronger Feature Regularity] For all $\mathbf{v} \in \mathbb{R}^{|\mathcal{S}|}$, $\|\Psi^\top \mathbf{v}\|_2 \leqslant C_\psi \|\mathbf{v}\|_\infty$, and $\|\Psi \mathbf{K}_\psi^{-1}\|_{2,\infty} \leqslant C'_\psi$, where $\|\mathbf{Y}\|_{2,\infty} = \max_i \sqrt{\sum_j \mathbf{Y}_{i,j}^2}$ is the $2, \infty$ norm (infinity norm over the $l_2$ norm of $\mathbf{Y}$'s columns).*

As it is explained in Yang & Wang (2020), this assumption can be satisfied when $\Psi$ is a set of sparse features or if $\Psi$ is a set of highly concentrated features.

The matrix estimator $\widetilde{\mathbf{M}}_n$ considered by Yang & Wang (2020) equals:

$$\widetilde{\mathbf{M}}_n = [\Sigma_n]^{-1} \sum_{n' < n, h \leqslant H} \phi_{n',h} \psi_{n',h}^\top \mathbf{K}_\psi^{-1}.$$

Where

$$\Sigma_n = \lambda \mathbb{I} + \sum_{n' < n, h \leqslant H} \phi_{n',h} \phi_{n',h}^\top$$

and $\Sigma_{n,h} = \Sigma_n + \sum_{h' < h} \phi_{n,h'} \phi_{n,h'}^\top$. It is easy to see that $\widetilde{\mathbf{M}}_n$ is the solution to the ridge regression problem:

$$\widetilde{\mathbf{M}}_n = \arg\min_{\mathbf{M}} \sum_{n' < n, h \leqslant H} \|\psi_{n',h}^\top \mathbf{K}_\psi^{-1} - \phi_{n',h}^\top \mathbf{M}\|_2^2 + \lambda \|\mathbf{M}\|_F^2. \tag{1}$$

It can be shown that with high probability and for all $t$ simultaneously all $\widetilde{\mathbf{M}}_n$ lie in a vicinity of $\mathbf{M}_\star$.

**Lemma 3.4.** *For all $\delta \in (0,1)$ with probability at least $1 - \delta$ for all $n \in \mathbb{N}$ simultaneously,*

$$\mathbf{M}_\star \in \{\mathbf{M} \in \mathbb{R}^{d \times d'} : \|(\Sigma_n)^{1/2}(\mathbf{M} - \widetilde{\mathbf{M}}_n)\|_{2,1} \leqslant d'\sqrt{\beta_n}\} := \mathbf{U}_n^{1,2}.$$
$$\mathbf{M}_\star \in \{\mathbf{M} \in \mathbb{R}^{d \times d'} : \|(\Sigma_n)^{1/2}(\mathbf{M} - \widetilde{\mathbf{M}}_n)\|_F \leqslant \sqrt{d'\beta_n}\} := \mathbf{U}_n^F.$$

*Where $\|\mathbf{B}\|_{2,1}$ denotes the $l_1$ norm of the $l_2$ norm of the columns of $\mathbf{B}$ while $\|\mathbf{B}\|_F$ corresponds to the Frobenius norm, $\sqrt{\beta_n} = R\sqrt{d \log\left(\frac{d' + d'nHL_\phi^2/\lambda}{\delta}\right)} + \sqrt{\lambda}S$ and $R = \|\mathbf{K}_\psi^{-1}\|L_\psi + SL_\phi$.*

The proof of Lemma 3.4 can be found in Appendix A. We can make use of Lemma 3.4 to show a regret guarantee for the MatrixRL algorithm from Yang & Wang (2020) (see Algorithm 1). Let's revisit the optimistic value function construction of the MatrixRL algorithm,

$$\forall (s,a) \in \mathcal{S} \times \mathcal{A}: \quad Q_{n,H+1}(s,a) = 0 \text{ and } \forall h \in [H]:$$
$$Q_{n,h}(s,a) = r(s,a) + \max_{\mathbf{M} \in \mathbf{U}_n^{1,2}} \phi(s,a)^\top \mathbf{M} \Psi^\top V_{n,h+1} \tag{2}$$

where

$$V_{n,h}(s) = \Pi_{[0,H]}\left[\max_a Q_{n,h}(s,a)\right] \quad \forall s, a, n, h.$$

$\Pi_{[0,H]}$ denotes the coordinate-wise clipping/projection operator onto the $[0,H]$ interval.

Let's define the "good" event of probability at least $1 - \delta$ where Lemma 3.4 holds as $\mathcal{E}$. We'll be making heavy use of the following 'determinant lemma',

---

**Algorithm 1** MatrixRL.

---

1: **Input:** An episodic MDP environment $\mathcal{M} = (\mathcal{S}, \mathcal{A}, P, s_0, r, H)$, features $\phi : \mathcal{S} \times \mathcal{A} \to \mathbb{R}^d$ and $\psi : \mathcal{S} \to \mathbb{R}^{d'}$, probability parameter $\delta \in (0, 1)$.
2: **Initialize:** $\Sigma_1 \leftarrow \mathbb{I} \in \mathbb{R}^{d \times d}$, $\mathbf{M}_1 \leftarrow \mathbf{0} \in \mathbb{R}^{d \times d'}$.
3: **for** episode $n = 1, \cdots, N$:
4:      Solve for $\widetilde{\mathbf{M}}_n$.
5:      Let $\{Q_{n,h}\}$ be given by Equation 2 using $\mathbf{U}_n^{1,2}$ and $\beta_n$ as in Lemma 3.4.
6:      **For** stage $h = 1, \cdots, H$:
7:          Let the current state be $s_{n,h}$ .
8:          Play action $a_{n,h} = \arg\max_{a \in \mathcal{A}} Q_{n,h}(s_{n,h}, a)$ .
9:          Record the next state $s_{n,h+1}$.
10: $\Sigma_{n+1} \leftarrow \Sigma_n + \sum_{h \leqslant H} \phi_{n,h} \phi_{n,h}^\top$.
11: Compute $\widetilde{\mathbf{M}}_{n+1}$ using (1).

---

**Lemma 3.5** (Determinant Lemma)*. (Lemma C.3 from (Pacchiano et al., 2020)) For any sequence of vectors* $\mathbf{x}_1, \ldots, \mathbf{x}_M \in \mathbb{R}^d$ *such that* $\|\mathbf{x}_q\|_2 \leqslant L$ *for all* $q \in [N]$. *Given a* $\lambda \geqslant 0$ *define* $\mathbf{D}_1 := \lambda \mathbf{I}$ *and for* $\ell \in \{2, \ldots, M+1\}$ *define* $\mathbf{D}_\ell := \lambda \mathbf{I} + \sum_{q=1}^{\ell-1} \mathbf{x}_q \mathbf{x}_q^\top$. *Then for all* $M \in \mathbb{N}$ *and* $b > 0$

$$\log\left(\frac{\det(\mathbf{D}_{M+1})}{\det(\lambda \mathbf{I})}\right) \leqslant d \log\left(1 + \frac{ML^2}{\lambda d}\right). \tag{3}$$

*and*

$$\sum_{q=1}^{M} \min\left\{b, \|\mathbf{x}_q\|_{\mathbf{D}_q^{-1}}^2\right\} \leqslant (1+b)d \log\left(1 + \frac{ML^2}{\lambda d}\right).$$

Our first result is to derive a sharper regret guarantee for the MatrixRL algorithm than in Yang & Wang (2020),

**Theorem 3.6.** *The regret satisfies,*

$$R(NH) \leqslant 8H\sqrt{NH \log\left(\frac{6 \log NH}{\delta}\right)} + 2\sqrt{2\gamma_N NHd \log\left(1 + \frac{NHL_\phi^2}{\lambda d}\right)} +$$

$$2L_\phi Hd\sqrt{\frac{\gamma_N}{\lambda}} \log\left(1 + \frac{NL_\phi^2}{\lambda d}\right)$$

*1. Under Assumption 3.2,*

$$\sqrt{\gamma_N} = 2C_\psi Hd'\sqrt{\beta_N} = 2C_\psi Hd'\left(R\sqrt{d \log\left(\frac{d' + d'NHL_\phi^2/\lambda}{\delta}\right)} + \sqrt{\lambda}S\right)$$

*2. Under the stronger Assumption 3.3,*

$$\sqrt{\gamma_N} = 2C_\psi H\sqrt{d'\beta_N} = 2C_\psi H\sqrt{d'}\left(R\sqrt{d \log\left(\frac{d' + d'NHL_\phi^2/\lambda}{\delta}\right)} + \sqrt{\lambda}S\right)$$

*with probability at least* $1 - 2\delta$.

The proof of Lemma 3.6 can be found in Appendix A.5. In contrast with the regret guarantees of Yang & Wang (2020), our bounds have a dependence on $H^{3/2}$ as opposed to $H^2$. We achieve this by using the following "lazy" version of the commonly used determinant lemma in the bandits/RL literature.

**Lemma 3.7.** *Let* $\mathbf{x}_{n,h} \in \mathbb{R}^{\tilde{d}}$ *satisfying* $\|\mathbf{x}_{n,h}\| \leqslant L$ *for some* $\tilde{d} \in \mathbb{N}$ *and let* $\mathbf{D}_{n,h} \in \mathbb{R}^{\tilde{d} \times \tilde{d}}$ *be a family of positive semidefinite matrices for* $n \in \mathbb{N}$ *and* $1 \leqslant h \leqslant H$ *such that* $\lambda \mathbf{I} \preceq \mathbf{D}_{n,h} \preceq \mathbf{D}_{n',h'}$ *if* $(n,h) \leqslant (n',h')$ *in the lexicographic order (i.e.* $n' > n$ *or* $h' \geqslant h$ *when* $n = n'$*). Define* $\mathbf{D}_n = \mathbf{D}_{n-1,H}$ *and* $\mathbf{D}_1 = \lambda \mathbf{I}$*. The following inequalities hold,*

$$\sum_{n=1}^{N} \sum_{h=1}^{H} \|\mathbf{x}_{n,h}\|_{\mathbf{D}_n^{-1}} \leqslant \sum_{n=1}^{N} \sum_{h=1}^{H} 2\|\mathbf{x}_{n,h}\|_{\mathbf{D}_{n,h}^{-1}} + \frac{2HL}{\sqrt{\lambda}} \log\left(\frac{\det(\mathbf{D}_{N+1})}{\det(\lambda \mathbf{I})}\right). \tag{4}$$

The proof of Lemma 3.7 can be found in Appendix A.1. As a corollary of Lemma 3.7,

**Corollary 3.8.** *The following inequalities hold,*

$$\sum_{n=1}^{N} \sum_{h=1}^{H} \|\phi_{n,h}\|_{\Sigma_n^{-1}} \leqslant \sum_{n=1}^{N} \sum_{h=1}^{H} 2\|\phi_{n,h}\|_{\Sigma_{n,h}^{-1}} + \frac{2L_\phi H d}{\sqrt{\lambda}} \log\left(1 + \frac{NHL_\phi^2}{\lambda d}\right). \tag{5}$$

The proof of Corollary 3.8 can be found in Appendix A.2. It allows us to transform a sum of inverse $\Sigma_n^{-1}$ norms to a sum of inverse $\Sigma_{n,h}^{-1}$ norms. This transformation comes at the cost of a 2 factor and a logarithmic cost with a $dH$ multiplier. Since it can be shown that $\sum_{n=1}^{N} \sum_{h=1}^{H} \|\phi_{n,h}\|_{\Sigma_{n,h}^{-1}} = \widetilde{\mathcal{O}}(\sqrt{dNH})$ where $\widetilde{\mathcal{O}}(\cdot)$ hides logarithmic factors, we conclude that $\sum_{n=1}^{N} \sum_{h=1}^{H} \|\phi_{n,h}\|_{\Sigma_n^{-1}} = \widetilde{\mathcal{O}}(\sqrt{dNH})$. This allows us to save a $\sqrt{H}$ factor in our final regret bound. Lemma 3.7 and Corollary 3.8 can be applied to any episodic linear setting and can be used to shave off a $\sqrt{H}$ factor form other episodic stationary linear models beyond MatrixRL.

## 4  Shared Structure Model

In this work we are concerned with understanding conditions under which sequential learning can be made more sample-efficient when simultaneously training in the presence of several related tasks. In contrast with other works that are concerned with the problem of learning from a set of related source tasks before engaging with a new target task, we are interested in understanding what benefits can be derived simultaneously from joint representation training across multiple RL problems. We borrow the subspace sharedness model from Yang et al. (2020) and generalize it from the setting of linear bandits to the previously described MatrixRL setting. We begin by assuming the learner has access to $P$ tasks encoded by the matrices $\{\mathbf{M}_\star^{(p)}\}_{p=1}^{P}$ with known reward functions $\{r^{(p)}\}_{p=1}^{P}$. We make the assumption the transitions factorize as $\mathbf{M}_\star^{(p)} = \mathbf{B}_\star \mathbf{A}_\star^{(p)}$ where $\mathbf{B}_\star \in \mathbb{R}^{d \times r}$ is a projection operator[1] and $\mathbf{A}_\star^{(p)} \in \mathbb{R}^{r \times d'}$. We require all of the matrices $\mathbf{M}_\star^{(p)}$ to satisfy Assumption 3.1, so $\|\mathbf{M}_\star^{(p)}\|_F = \|\mathbf{A}_\star^{(p)}\|_F \leqslant \sqrt{d'}S$.

We are interested in designing an algorithm that bounds the "shared regret", defined as

$$R_P(NH) = \sum_{n=1}^{N} \sum_{p=1}^{P} V_1^{\pi_\star^{(p)}}(s_{n,1}^{(p)}) - V_1^{\pi_n^{(p)}}(s_{n,1}^{(p)}),$$

where $s_{n,1}^{(p)}$ is the starting state for task $p$ in epsiode $n$, $\pi_n^{(p)}$ is the policy used by task $p$ during epsiode $n$, and $\pi_\star^{(p)}$ is the optimal policy of task $p$. Notice that instead of optimizing the usual form of the single task regret, here we are interested in minimizing the aggregate regret incurred across all tasks. The learner's objective is to leverage the shared structure among the tasks to incur a regret $R_P(NH)$ smaller then what is obtained by learning each task in isolation–a shared regret equal to $P$ times the single-task MatrixRL regret upper bound.

In this framework, the transition dynamics across MDPs are coupled because the agent's feature embedding of state-action pairs lie in a common low-dimensional subspace. If the learner had knowledge of $\mathbf{B}_\star$, they would be able to use projected features of the form $\tilde{\phi}(s,a) = \mathbf{B}_\star \phi(s,a)$ in their exploration. This would allow the learner to incur regret scaling only in $r$, independently of $d$. Although it is impossible to completely eliminate the $d$-dependence without apriori knowledge of $\mathbf{B}_\star$, we show that in some cases it is possible to improve the $d$-dependence. Our main result can be summarized as follows,

---

[1]Recall that a linear operator $\mathbf{P}$ is a projection if $\mathbf{P}^2 \mathbf{v} = \mathbf{P}\mathbf{v}$.

**Theorem 4.1** (Informal)**.** *There exists an algorithm for joint learning over a set of related tasks* $\{\mathbf{M}_\star^{(p)} = \mathbf{B}_\star \mathbf{A}_\star^{(p)}\}_{p \in [P]}$ *that achieves a regret of*

$$R_P(NH) = \widetilde{\mathcal{O}}\left(\left(Hd\sqrt{rP} + HP\sqrt{rd}\right)\sqrt{NH}\right),$$

*with high probability, where* $\widetilde{\mathcal{O}}$ *hides logarithmic factors*

Recall that for an isolated task in order to recover an estimator $\widetilde{\mathbf{M}}$ of $\mathbf{M}_\star$ given $n-1$ trajectories of horizon $H$ we solve $d'$ independent ridge regression problems (one per column) as defined by Equation 1.

In the multi-task setting with shared structure, we instead consider the following quadratic objective that weaves together the estimation of the task-specific $\{\mathbf{A}^p\}_{p=1}^P$ parameters with that of the shared $\mathbf{B}$ projection matrix.[2]

$$\underset{\substack{\mathbf{B} \in \mathcal{P}_{d,r}, \\ \|\mathbf{A}^{(1)}\|_F \leqslant \sqrt{d'}S, \cdots, \|\mathbf{A}^{(P)}\|_F \leqslant \sqrt{d'}S}}{\arg\min} \quad F(\mathbf{B}, \mathbf{A}^{(1)}, \cdots, \mathbf{A}^{(P)}) \tag{6}$$

$$F(\mathbf{B}, \mathbf{A}^{(1)}, \cdots, \mathbf{A}^{(P)}) = \sum_{p \in [P]} \lambda \|\mathbf{A}^{(p)}\|_F^2 + \sum_{n' < n, h \leqslant H} \left\| \left(\psi_{n',h}^{(p)}\right)^\top \mathbf{K}_\psi^{-1} - \left(\phi_{n',h}^{(p)}\right)^\top \left(\mathbf{B}\mathbf{A}^{(p)}\right) \right\|_2^2$$

Where $\mathcal{P}_{d,r}$ corresponds to the set of all $d \times r$ projection matrices with $r$ orthonormal columns and the search space for $\mathbf{A}^{(p)}$ is the Frobenius ball of radius $\sqrt{d'}S$ in the space of matrices $\mathbb{R}^{r \times d'}$.

Notice that by virtue of the orthogonality of $\mathbf{B}$'s columns (i.e. $\mathbf{B}^\top \mathbf{B} = \mathbb{I}_r$ ) the regularizer satisfies $\|\mathbf{B}\mathbf{A}^{(p)}\|_F^2 = \|\mathbf{A}^{(p)}\|_F^2$. We use the notation $\widetilde{\mathbf{B}}_n, \widetilde{\mathbf{A}}_n^{(1)}, \cdots, \widetilde{\mathbf{A}}_n^{(P)}$ to refer to the resulting estimators for the shared projection matrix and the low rank dynamics matrices for each of the tasks $p = 1, \cdots, P$ right before the $n$th batch of $P$ trajectories is collected.

We start by proving a series of data dependent bounds on the estimates $\widetilde{\mathbf{B}}, \widetilde{\mathbf{A}}_n^{(1)}, \cdots, \widetilde{\mathbf{A}}_n^{(P)}$ that will serve as the analogous shared-structure versions of Lemma 3.4.

Now we show a bound for the data-dependent distance between $\widetilde{\mathbf{B}}_n, \widetilde{\mathbf{A}}_n^{(1)}, \cdots, \widetilde{\mathbf{A}}_n^{(P)}$ and the true parameters $\mathbf{B}_\star, \mathbf{A}_\star^{(1)}, \cdots, \mathbf{A}_\star^{(P)}$.

**Lemma 4.2.** *For any* $\delta \in (0,1)$ *the following bound holds,*

$$\sum_{p \in [P]} \lambda \left\| \widetilde{\mathbf{A}}_n^{(p)} \right\|_F^2 + \frac{1}{2} \left\| \left(\Sigma_n^{(p)}\right)^{1/2} \left(\mathbf{B}_\star \mathbf{A}_\star^{(p)} - \widetilde{\mathbf{B}}_n \widetilde{\mathbf{A}}_n^{(p)}\right) \right\|_F^2 \leqslant \beta'_{nH}(\delta) + \sum_{p \in [P]} \lambda \|\mathbf{A}_\star^{(p)}\|_F^2$$

*with probability at least* $1 - \delta$ *for all* $n \in \mathbb{N}$ *and where*

$$\beta'_{nH}(\delta) = 1 + L_\phi S + \frac{b^2}{2R^2} +$$

$$(12R^2 + b)\left(2\log\log\left(2\left(nHP\right)\right) + 3 + \log\frac{1}{\delta} + (dr + rd'P)\left(\log(5S) + \log nHP + \log 2RL_\phi\right)\right)$$

*And* $b = 2Rd'SL_\psi$.

The proof of Lemma 4.2 can be found in Appendix B.1. In contrast with the results of Lemma 3.4, the guarantees of Lemma 4.2 apply to the sum of the errors across all $P$ tasks. As we'll see in the coming discussion this is the main source of difficulties in designing a reinforcement learning algorithm that successfully makes use of this result to construct optimistic value functions. We can use Lemma 4.2 to obtain the following high probability confidence interval jointly around $\widetilde{\mathbf{B}}_n$ and $\{\widetilde{\mathbf{A}}_n^{(p)}\}_{p=1}^P$, which is one of our main results:

---

[2]Our results will also be true when the $\psi, \phi$ maps are task-dependent. In this case, the only change to our results would require making $\mathbf{K}_\psi$ task-dependent.

**Lemma 4.3.** *For any $\delta \in (0,1)$ with probability at least $1 - \delta$ for all $n \in \mathbb{N}$ simultaneously,*

$$\left\{ \mathbf{M}_\star^{(p)} = \mathbf{B}_\star \mathbf{A}_\star^{(p)} \right\}_{p=1}^P \in \left\{ \{\mathbf{B}\mathbf{A}^{(p)}\}_{p=1}^P \ s.t. \ \sum_p \left\| \left( \Sigma_n^{(p)} \right)^{1/2} \left( \mathbf{B}\mathbf{A}^{(p)} - \tilde{\mathbf{B}}_n \tilde{\mathbf{A}}_n^{(p)} \right) \right\|_F^2 \leqslant \gamma_n(\delta) \right\}$$

$$\subseteq \underbrace{\left\{ \{\mathbf{M}^{(p)}\}_{p=1}^P \ s.t. \ \sum_p \left\| \left( \Sigma_n^{(p)} \right)^{1/2} \left( \mathbf{M}^{(p)} - \tilde{\mathbf{B}}_n \tilde{\mathbf{A}}_n^{(p)} \right) \right\|_F^2 \leqslant \gamma_n(\delta) \right\}}_{:= \tilde{\mathbf{U}}_n^F(\delta)}$$

*where $\gamma_n(\delta) = 2\beta_n'(\delta) + 2P\sqrt{d'}S\lambda$ and $\beta_n'$ is defined as in Lemma 4.2.*

*Proof.* Lemma 4.2 implies that with probability at least $1 - \delta$ for all $n \in \mathbb{N}$,

$$\sum_{p \in [P]} \lambda \left\| \tilde{\mathbf{A}}_n^{(p)} \right\|_F^2 + \frac{1}{2} \left\| \left( \Sigma_n^{(p)} \right)^{1/2} \left( \mathbf{B}_\star \mathbf{A}_\star^{(p)} - \tilde{\mathbf{B}}_n \tilde{\mathbf{A}}_n^{(p)} \right) \right\|_F^2 \leqslant \beta_{nH}'(\delta) + \sum_{p \in [P]} \lambda \|\mathbf{A}_\star^{(p)}\|_F^2$$

Since $\|\mathbf{A}_\star^{(p)}\|_F \leqslant \sqrt{d'}S$, this implies that

$$\sum_{p \in [P]} \left\| \left( \Sigma_n^{(p)} \right)^{1/2} \left( \mathbf{B}_\star \mathbf{A}_\star^{(p)} - \tilde{\mathbf{B}}_n \tilde{\mathbf{A}}_n^{(p)} \right) \right\|_F^2 \leqslant 2\beta_{nH}'(\delta) + 2P\sqrt{d'}S\lambda.$$

The result follows. □

From here on we use the name $\mathcal{E}'$ to define the event of Lemma 4.3 where the sum of the square of the confidence intervals across all tasks is bounded by $\gamma_n(\delta)$. Lemma 4.3 implies $\mathbb{P}(\mathcal{E}') \geqslant 1 - \delta$.

---

**Algorithm 2** Shared-MatrixRL.

1: **Input:** Episodic MDP environments $\{\mathcal{M}^{(p)}\}_{p \in [P]} = (\mathcal{S}, \mathcal{A}, \mathbb{P}^{(p)}, s_0, r, H)$, features $\phi^{(p)} : \mathcal{S} \times \mathcal{A} \to \mathbb{R}^d$ and $\psi^{(p)} : \mathcal{S} \to \mathbb{R}^{d'}$, probability parameter $\delta \in (0,1)$.

2: **Initialize:** $\{\Sigma_1^{(p)} \leftarrow \mathbb{I} \in \mathbb{R}^{d \times d}\}_{p=1}^P$, $\{\mathbf{M}_1^{(p)} \leftarrow \mathbf{0} \in \mathbb{R}^{d \times d'}\}$.

3: **For** episode $n = 1, \cdots, N$:

4:     Solve Problem 6 and compute $\tilde{\mathbf{B}}_n, \tilde{\mathbf{A}}_n^{(1)}, \cdots, \tilde{\mathbf{A}}_n^{(P)}$.

5:     Let $\{Q_{n,h}^{(p)}\}_{p=1}^P$ be given by $Q_{n,h}^{(p)}(s,a) = Q_{n,h}^{(p)}(s, a, \{\overline{\mathbf{M}}_n^{(p)}\}_{p=1}^P)$

6:     where,

$$\{\overline{\mathbf{M}}_n^{(p)}\}_{p=1}^P = \underset{\{\mathbf{M}^{(p)}\}_{p=1}^P \in \tilde{\mathbf{U}}_n^F(\delta)}{\arg\max} \sum_p V_{n,1}^{(p)}(s_{n,1}^{(p)}, \{\mathbf{M}^{(p)}\}_{p=1}^P). \tag{7}$$

7:     Where $\{s_{n,1}^{(p)}\}_{p=1}$ is the set of first states seen at the start of their episodes by all tasks.

8:     **For** $p = 1, \cdots, P$:

9:         **For** stage $h = 1, \cdots, H$ :

10:             Let the current state be $s_{n,h}^{(p)}$.

11:             Play action $a_{n,h}^{(p)} = \arg\max_{a \in \mathcal{A}} Q_{n,h}^{(p)}(s_{n,h}^{(p)}, a)$.

12:             Record the next state $s_{n,h+1}^{(p)}$.

13:         Update $\Sigma_{n+1}^{(p)} \leftarrow \Sigma_n^{(p)} + \sum_{h \leqslant H} \left( \phi_{n,h}^{(p)} \right) \left( \phi_{n,h}^{(p)} \right)^\top$ for all $p \in [P]$.

---

We now introduce the Shared-MatrixRL algorithm. In contrast with the simple MatrixRL in Algorithm 1, Shared-MatrixRL makes use of a shared confidence interval for the $\tilde{\mathbf{B}}_n, \{\tilde{\mathbf{A}}_n^{(p)}\}_{p=1}^P$ matrices. We define the following optimistic $Q-$functions for the task family,

$$\forall \{\mathbf{M}^{(p)}\}_{p \in [P]} \text{ and } \forall (s,a) \in \mathcal{S} \times \mathcal{A}: \quad Q_{n,H+1}^{(p)}(s, a, \{\mathbf{M}^{(p)}\}_{p=1}^P) = 0 \quad \forall p \in [P] \text{and } \forall h \in [H]:$$

$$Q_{n,h}^{(p)}(s, a, \{\mathbf{M}^{(p)}\}_{p \in [P]}) = r^{(p)}(s_p, a_p) + \phi^{(p)}(s_p, a_p)^\top \mathbf{M}^{(p)} \left( \mathbf{\Psi}^{(p)} \right)^\top V_{n,h+1}^{(p)}(\{\mathbf{M}^{(p)}\}_{p=1}^P)$$

where $V_{n,h+1}^{(p)}(\{\mathbf{M}^{(p)}\}_{p=1}^{P})$ is a vector of dimension $|\mathcal{S}|$ corresponding to the value functions of task $p$ under model $\mathbf{M}^{(p)}$. For all $s, a, n, h$,

$$V_{n,h}^{(p)}(s, \{\mathbf{M}^{(p)}\}_{p=1}^{P}) = \Pi_{[0,H]}\left[\max_a Q_{n,h}^{(p)}(s, a, \{\mathbf{M}^{(p)}\}_{p=1}^{P})\right].$$

The definition of the parametric $Q$ functions $Q_{n,h}^{(p)}(s, a, \{\mathbf{M}^{(p)}\}_{p=1}^{P})$ and value functions $V_{n,h}^{(p)}(s, \{\mathbf{M}^{(p)}\}_{p=1}^{P})$ is required to define the joint optimistic objective for the set of $P$ tasks of Equation 7. We define the optimistic value functions as,

$$Q_{n,h}^{(p)}(s, a) = Q_{n,h}^{(p)}(s, a, \{\overline{\mathbf{M}}_n^{(p)}\}_{p=1}^{P}), \qquad V_{n,h}^{(p)}(s) = V_{n,h}^{(p)}(s, \{\overline{\mathbf{M}}_n^{(p)}\}_{p=1}^{P})$$

The optimization problem of Equation 7 requires to solve for $\{\overline{\mathbf{M}}_n^{(p)}\}_{p=1}^{P}$ optimizes the sum of values as 'seen' from the initial states $\{s_{n,1}^{(p)}\}_{p\in[P]}$ of the $P$ tasks at the beginning of the $n$th episode. This form of optimism is required to ensure the constraint $\{\overline{\mathbf{M}}^{(p)}\}_{p=1}^{P} \subset \widetilde{\mathbf{U}}_n^F(\delta)$ is satisfied.

**Limitations.** Shared-MatrixRL works in a similar way to the single task Matrix RL algorithm; a policy is executed in each of the component tasks based on a series of optimistic $Q$ values. The data collected by the learner is then used to update the component models via Equation 6. The chief difference in our approach to the multi task setting lies in the definition of the shared $Q$ functions. This is what allows us to make use of the shared confidence interval of Lemma 4.3. Unfortunately this means the computation of the 'optimistic models' $\{\overline{\mathbf{M}}_n^{(p)}\}_{p=1}^{P}$ is intractable since it requires the computation and storage of the $Q$ values $Q_{n,h}^{(p)}(s, a, \{\mathbf{M}_n^{(p)}\}_{p=1}^{P})$ for all feasible values of $\{\mathbf{M}_n^{(p)}\}_{p\in[P]}$ and then solve for $\{\overline{\mathbf{M}}_n^{(p)}\}_{p=1}^{P}$. This situation is not as severe as it seems since the computation of the optimistic $Q$ functions in the original MatrixRL algorithm (and even in the OFUL algorithm for linear bandits Abbasi-Yadkori et al. (2011)) is also an intractable problem. Another potential drawback of Algorithm 2 is its requirement to have knowledge of the initial states $\{s_{n,1}^{(p)}\}_{p=1}^{P}$. An astute reader may posit it to be possible to overcome this issue by using Thompson Sampling Agrawal & Goyal (2013); Abeille & Lazaric (2017). In this case we would sample a set of models $\{\overline{\mathbf{M}}_n^{(p)}\}_{p=1}^{P}$ from block gaussian distribution where each block is centered around each $\widetilde{\mathbf{M}}_n^{(p)}$. Sampling from this posterior does not require knowledge of $\{s_{n,1}^{(p)}\}_{p=1}^{P}$. Unfortunately, this strategy would cause the degradation of the regret upper bound to a level that is not competitive with the strategy of solving each task independently. We leave the removal of the assumption on $\{s_{n,1}^{(p)}\}_{p=1}^{P}$ as future work.

In order to prove the Shared-MatrixRL satisfies a satisfactory sublinear regret guarantee we start by showing optimism holds for the shared representations parameterized by $\{\overline{\mathbf{M}}_n^{(p)}\}_{p=1}^{P}$.

**Lemma 4.4** (Optimism). *Whenever $\mathcal{E}'$ holds,*

$$\sum_{p\in[P]} V_1^{\pi_\star^{(p)}}(s_{n,1}^{(p)}) \leqslant \sum_{p\in[P]} V_{n,1}^{(p)}(s_{n,1}^{(p)}).$$

*Proof.* Since

$$V_{n,1}^{(p)}(s_{n,1}^{(p)}) = V_{n,1}\left(s_{n,1}^{(p)}, \{\overline{\mathbf{M}}_n^{(p)}\}_{p=1}^{P}\right)$$

the definition of $\{\overline{\mathbf{M}}_n^{(p)}\}_{p=1}^{P}$ implies that,

$$\sum_{p\in[P]} V_{n,1}(s_{n,1}^{(p)}, \{\overline{\mathbf{M}}_n^{(p)}\}_{p=1}^{P}) \geqslant \sum_{p\in[P]} V_{n,1}(s_{n,1}^{(p)}, \{\mathbf{B}_\star \mathbf{A}_\star^{(p)}\}_{p=1}^{P})$$

Since $V_{n,1}(s_{n,1}^{(p)}, \mathbf{B}_\star, \{\mathbf{A}_\star^{(p)}\}_{p=1}^{P}) = V_1^{\pi_\star^{(p)}}(s_{n,1}^{(p)})$, the result follows. □

Similarly we can use our confidence interval bounds to prove the following bound on the bellman error.

**Lemma 4.5.** *If Assumption 3.3 holds and $\mathcal{E}'$ is true then for $h \in [H]$,*

$$\sum_{p \in [P]} Q_{n,h}^{(p)}(s_{n,h}^{(p)}, a_{n,h}^{(p)}) - \left( r(s_{n,h}^{(p)}, a_{n,h}^{(p)}) + \mathbb{P}^{(p)}(\cdot | s_{n,h}^{(p)}, a_{n,h}^{(p)})^\top V_{n,h+1}^{(p)} \right)$$

$$\leqslant 2 C_\psi H \sqrt{\gamma_n(\delta) \sum_{p \in [P]} \|\phi_{n,h}^{(p)}\|_{\left(\Sigma_n^{(p)}\right)^{-1}}^2}$$

The proof of Lemma 4.5 can be found in Appendix B.2. Having established that optimism holds, we can use a similar set of techniques as in the proof of Theorem 3.6 to show a regret guarantee. First we derive Corollary 4.6, an equivalent version to Corollary 3.8. This allows us to maintain the $\sqrt{H}$ factor improvement in the multitask setting. This result is a consequence of Lemma 3.7.

**Corollary 4.6.** *The following inequalities hold,*

$$\sum_{n=1}^N \sum_{h=1}^H \sqrt{\sum_{p \in [P]} \|\phi_{n,h}^{(p)}\|_{\left(\Sigma_n^{(p)}\right)^{-1}}^2} \leqslant \sum_{n=1}^N \sum_{h=1}^H 2 \sqrt{\sum_{p \in [P]} \|\phi_{n,h}^{(p)}\|_{\left(\Sigma_{n,h}^{(p)}\right)^{-1}}^2} + \frac{2 L_\phi H dP}{\sqrt{\lambda}} \log\left(1 + \frac{NHL_\phi^2}{\lambda d}\right). \quad (8)$$

*Proof.* Define $NHP$ variables $\mathbf{x}_{n,h,p} \in \mathbb{R}^{dP}$ ordered lexicographically and satisfying $\mathbf{x}_{n,h} = (\phi_{n,h}^{(1)}, \cdots, \phi_{n,h}^{(P)})$ where $\phi_{n,h}^{(p)}$ is located in the $p-$th $d$ dimensional slot of $\mathbf{x}_{n,h}$ for all $p \in [P]$. In this case, $\mathbf{D}_{n,h}$ is a block diagonal matrix (with $d \times d$ diagonal blocks equal to $\Sigma_{n,h}$) such that $\|\mathbf{x}_{n,h}\|_{\mathbf{D}_{n,h}^{-1}} = \sqrt{\sum_{p \in [P]} \|\phi_{n,h}^{(p)}\|_{\left(\Sigma_{n,h}^{(p)}\right)^{-1}}^2}$. By definition $\|\mathbf{x}_{n,h}\|_{\mathbf{D}_n^{-1}} = \sqrt{\sum_{p \in [P]} \|\phi_{n,h}^{(p)}\|_{\left(\Sigma_n^{(p)}\right)^{-1}}^2}$. As a consequence of Lemma 3.7,

$$\sum_{n=1}^N \sum_{h=1}^H \sqrt{\sum_{p \in [P]} \|\phi_{n,h}^{(p)}\|_{\left(\Sigma_n^{(p)}\right)^{-1}}^2} \leqslant \sum_{n=1}^N \sum_{h=1}^H 2 \sqrt{\sum_{p \in [P]} \|\phi_{n,h}^{(p)}\|_{\left(\Sigma_{n,h}^{(p)}\right)^{-1}}^2} + \frac{2 L_\phi H}{\sqrt{\lambda}} \log\left(\frac{\det(\mathbf{D}_{N+1})}{\det(\lambda \mathbf{I}_{dP})}\right).$$

Where we have used the notation $\mathbf{I}_s$ to denote the $s \times s$ dimensional identity matrix. By definition of $\mathbf{D}_{N+1}$ we see that $\det(\mathbf{D}_{N+1}) = \prod_{p=1}^P \det(\Sigma_{N+1}^{(p)})$ and therefore,

$$\log\left(\frac{\det(\mathbf{D}_{N+1})}{\det(\lambda \mathbf{I}_{dP})}\right) = \sum_{p=1}^P \log\left(\frac{\det(\Sigma_{N+1}^{(p)})}{\det(\lambda \mathbf{I}_d)}\right) \leqslant Pd \log\left(1 + \frac{NHL_\phi^2}{\lambda d}\right).$$

Where the last inequality follows from Equation 3 in Lemma 3.5. The result follows. $\qquad \square$

Similar to Corollary 3.8, the result of Corollary 4.6 allows us to transform inverse norms defined by the matrices $\left(\Sigma_n^{(p)}\right)^{-1}$, into inverse norms defined by the matrices $\left(\Sigma_{n,h}^{(p)}\right)^{-1}$, at a constant multiplicative cost plus a logarithmic term with a $dHP$ multiplier.

**Theorem 4.7.** *The regret of Shared-MatrixRL satisfies,*

$$R_P(NH) \leqslant H\sqrt{NHP \log\left(\frac{6 \log NH}{\delta}\right)} + 4 C_\psi H^2 dP \left(1 + \frac{L_\phi^2}{\sqrt{\lambda}}\right) \log\left(1 + \frac{NHL_\phi^2}{\lambda d}\right) \sqrt{\gamma_N(\delta)} +$$

$$2 C_\psi H \sqrt{\gamma_N(\delta) NHPd \left(1 + \frac{L_\phi^2}{\sqrt{\lambda}}\right) \log\left(1 + \frac{NHL_\phi^2}{\lambda d}\right)}.$$

*With probability at least $1 - 2\delta$.*

The proof can be found in Appendix B. Since $\gamma_N(\delta) \approx dr + rP$ (up to logarithmic factors and ignoring polynomial dependencies on $d'$) Theorem 4.7 implies,

**Corollary 4.8.** *The regret of Algorithm 2 satisfies,*

$$R_P(NH) \leqslant \tilde{\mathcal{O}}\left(H\sqrt{NHP} + H\sqrt{dr + rP}\sqrt{NHPd}\right) = \tilde{\mathcal{O}}\left(\left(Hd\sqrt{rP} + HP\sqrt{rd}\right)\sqrt{NH}\right).$$

*With probability at least $1 - 2\delta$.*

This result improves upon the shared regret of order $\tilde{\mathcal{O}}(HdP\sqrt{NH})$ achieved by using the MatrixRL algorithm to learn each task independently. Interestingly, learning the tasks' shared structure only becomes beneficial when $r \ll d$ and $r \ll P$. To explain this phenomenon observe that the degrees of freedom (i.e. the number of parameters to learn) in Shared-MatrixRL equals $dr + Pr$. The degrees of freedom of running $P$ independent copies of MatrixRL in contrast equals $dPd'$. For shared representation learning to be more efficient than learning each task alone, we require $dr + Prd' \ll dPd'$. This is why for Shared-MatrixRL learning to be truly beneficial (and attain a smaller regret upper bound than running $P$ tasks independently) we require $dr \ll dPd'$ and $Prd' \ll dPd'$. For example when the number of tasks is small and $P \ll r$, learning the shared matrix $\mathbf{B}_\star$ may require more data than learning the $dPd'$ parameters of estimating the models for all $P$ tasks independently. Although we have not developed a lower bound for the specific MatrixRL setting, the results of Yang et al. (2020) provide evidence to posit the regret upper bound for Shared-MatrixRL in Theorem 4.7 is optimal.

## 5 Computationally Efficient Shared-MatrixRL

Algorithm 2 has two computationally intensive components. First, solving for $\widetilde{\mathbf{B}}_n, \widetilde{\mathbf{A}}_n^{(1)}, \cdots, \widetilde{\mathbf{A}}_n^{(P)}$ and second, solving for Equation 7. The first objective may be difficult to solve because it involves solving a bilinear quadratic optimization problem. The second one can prove even more challenging first because it requires a way to 'store' the parametric value functions $V_{n,1}^{(p)}(s, \{\mathbf{M}^{(p)}\}_{p=1}^P)$ (these functions may be highly non-linear), and second because solving for Equation 7 involves optimizing a non-convex objective.

In this section we show that, given access to a computational oracle for Problem 6 and assuming $\mathcal{S}, \mathcal{A}$ are finite, there exists a computationally efficient procedure for solving for the joint optimistic objective of Equation 7 of Algorithm 2. As it is mentioned in the discussion surrounding Equation 7 of Yang & Wang (2020), the confidence bonus of Equation 2 can be substituted by

$$Q_{n,h}(s,a) = r(s,a) + \phi(s,a)^\top \widetilde{\mathbf{M}}_n \mathbf{\Psi}^\top V_{n,h+1} + 2L_\Psi H\sqrt{\beta_n}\|\phi(s,a)\|_{\Sigma_n^{-1}}$$

This corresponds to explicitly solving for the optimistic model maximizing the $Q$ values at state action pair $(s,a)$ and in-episode time $h$. Let $\tau^{(p)}$ be a set of $P$ confidence radii. In the multi-task setting, let's consider enforcing,

$$\overline{\mathbf{M}}_n^{(p)} \in \left\{ \left\|\left(\Sigma_n^{(p)}\right)^{1/2}\left(\mathbf{M}^{(p)} - \widetilde{\mathbf{B}}_n\widetilde{\mathbf{A}}_n^{(p)}\right)\right\|_F \leqslant \tau^{(p)} \right\} := \widetilde{\mathbf{U}}_n^F(\delta, p, \tau^{(p)})$$

If $\sum_{p=1}^P (\tau^{(p)})^2 \leqslant \gamma_n(\delta)$, we can allow for the per-state maximization of the optimistic models as in the single task setting (see Equation 2) and obviate solving for problem 7 in Algorithm 2. If we call $\overline{\mathbf{M}}_n^{(p)}(s,a)$ the model in $\mathbf{U}_n^F(\delta, p, \tau^{(p)})$ achieving the argmax in the definition $Q_{n,h}^{(p)}(s, a, \tau^{(p)}) = r(s,a) + \max_{\mathbf{M} \in \mathbf{U}_n^F(\delta, p, \tau^{(p)})} \phi(s,a)^\top \mathbf{M}\mathbf{\Psi}^\top V_{n,h+1}^{(p)}(\tau^{(p)})$. This is because restricting the individual confidence radii for model $p$ to be upper bounded by $\tau^{(p)}$ for all state action pairs ensures that,

$$\sum_{p \in [P]} Q_{n,h}^{(p)}(s_{n,h}^{(p)}, a_{n,h}^{(p)}, \tau^{(p)}) - \left(r(s_{n,h}^{(p)}, a_{n,h}^{(p)}) + \mathbb{P}^{(p)}(\cdot|s_{n,h}, a_{n,h})^\top V_{n,h+1}^{(p)}(\tau^{(p)})\right)$$

$$\leqslant \sum_{p \in [P]} \left\|\left(\phi_{n,h}^{(p)}\right)^\top\left(\overline{\mathbf{M}}_n^{(p)} - \mathbf{M}_\star^{(p)}\right)\right\|_2 \left\|\left(\mathbf{\Psi}^{(p)}\right)^\top V_{n,h+1}^{(p)}(\tau^{(p)})\right\|_2$$

$$\leqslant \sum_{p \in [P]} C_\psi \left\|V_{n,h+1}^{(p)}(\tau^{(p)})\right\|_\infty \left\|\left(\phi_{n,h}^{(p)}\right)^\top\left(\overline{\mathbf{M}}_n^{(p)} - \mathbf{M}_\star^{(p)}\right)\right\|_2$$

If $\{\tau^{(p)}\}_{p=1}^{P}$ are defined such that $\sum_{p=1}^{P}\left(\tau^{(p)}\right)^2 \leqslant \gamma_n(\delta)$ the same arguments as in the proof of Lemma 4.5 imply,

$$\sum_{p\in[P]} Q_{n,h}^{(p)}(s_{n,h}^{(p)}, a_{n,h}^{(p)}, \tau^{(p)}) - \left(r(s_{n,h}^{(p)}, a_{n,h}^{(p)}) + \mathbb{P}^{(p)}(\cdot|s_{n,h}, a_{n,h})^\top V_{n,h+1}^{(p)}(\tau^{(p)})\right)$$

$$\leqslant 2C_\psi H \sqrt{\gamma_n(\delta) \sum_{p\in[P]} \|\phi_{n,h}^{(p)}\|_{\left(\Sigma_n^{(p)}\right)^{-1}}^2}$$

The $Q$ functions $Q_{n,h}^{(p)}(\cdot, \cdot, \tau^{(p)})$ satisfy,

$$Q_{n,h}^{(p)}(s, a, \tau^{(p)}) = r^{(p)}(s, a) + \phi^{(p)}(s, a)^\top \widetilde{\mathbf{B}}_n \widetilde{\mathbf{A}}_n^{(p)} \left(\mathbf{\Psi}^{(p)}\right)^\top V_{n,h+1}^{(p)}(\tau^{(p)}) +$$

$$2L_\Psi H \tau^{(p)} \|\phi^{(p)}(s, a)\|_{\left(\Sigma_n^{(p)}\right)^{-1}}$$

Where

$$V_{n,h+1}^{(p)}(\tau^{(p)}) = \Pi_{[0,H]} \left[\max_a Q_{n,h}^{(p)}(s, a, \tau^{(p)})\right] \quad \forall s, a, n, h.$$

If $\mathcal{S}, \mathcal{A}$ are finite sets, then for any fixed set of thresholds $\{\tau^{(p)}\}$, solving for $Q^{(p)}(s, a, \tau^{(p)})$ can be expressed as the solution to a linear program in the variables $V_{n,h+1}^{(p)}$ and $Q_{n,h}^{(p)}$. By adding a quadratic constraint of the form $\sum_{p\in[P]}\left(\tau^{(p)}\right)^2 \leqslant \gamma_n(\delta)$ the resulting optimization problem over all tasks $p \in [P]$ becomes the convex Quadratically Constrained Linear Program (QCLP),

$$\max_{p\in[P]} \sum_{p=1}^{P} V_{n,1}^{(p)}(s_{n,1}^{(p)}, \tau^{(p)}) \text{ s.t. } \sum_{p\in[P]}\left(\tau^{(p)}\right)^2 \leqslant \gamma_n(\delta),$$

and thus it will take poly $\left(\frac{1}{NH}\right)$ operations to arrive at an $\frac{1}{N^2H^2}$ approximate solution for this problem. This is enough to guarantee optimism up to an overall error of order $\frac{1}{NH}$. See the discussion in Chapter 4 of Boyd et al. (2004) on how to solve QCLP problems efficiently.

## 6 Conclusion

In this work we are the first to analyze the problem of joint training across a set of related Markov Decision Processes. We show that when the training tasks' transition dynamics can be embedded in a common low-dimensional subspace of dimension $r$, a joint training algorithm can obtain regret $\widetilde{\mathcal{O}}\left(\left(Hd\sqrt{rP} + HP\sqrt{rd}\right)\sqrt{NH}\right)$ as opposed to $\widetilde{\mathcal{O}}(HdP\sqrt{NH})$ – the regret of learning each task separately ignoring the shared task structure. Our training method solves a quadratic optimization problem that jointly penalizes the shared and task-dependent model parameters (see Equation 6). We expect the techniques we have introduced in this work, including the multitask least squares objective of Equation 6 and the parametric $Q$ functions $Q_{n,h}^{(p)}(s, a, \{\mathbf{M}^{(p)}\}_{p\in[P]})$, to have applications in other MDP models with function approximation–such as Linear MDPs Jun et al. (2019); Zhou et al. (2021) amongst others.

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
