# OpenReview forum: "Joint Representation Training in Sequential Tasks with Shared Structure"
_TMLR — Rejected by TMLR_

### Review · Reviewer_aEHw · 2022-09-26

**Summary Of Contributions:**

The paper studies how to utilize the shared structures of multiple reinforcement learning environments to achieve better statistical efficiency. They use the matrix RL model proposed in [Yang & Wang 2019] and generalize that to multiple tasks, which share a factor of the core matrices. The paper proposes an approach to learning in these environments simultaneously and solves the optimistic policy for each of these environments with the knowledge of the shared structures. The proposed algorithm takes a non-convex optimization form and resolves difficulties in achieving optimism in the obtained policies. The final regret bound receives a saving in terms of the ambient dimension d. The paper additionally discusses the computational issues as well as the limitations of the algorithm.

**Requested Changes:**

As stated in the weakness section, it would be better to:
* revise the preliminary section to make it more concise
* give a technical overview in the introduction section -- what are the challenges of generalizing the single-task matrixRL algorithm
* discuss why the simultaneous acting model is significant and relevant (e.g., both theoretically and practically)
* discuss or provide a lower bound
* correct typos, e.g.,  -- sec 4, paragraph 2: "smaller then"
* it would be better to discuss the bounds immediately after the presentation of Thm 4.1: why this bound makes sense.

**Strengths And Weaknesses:**

Strengths:
- The paper proposes a method of how to utilize the shared structures in multiple tasks, and how saving can be achieved.
- The handling of the non-convex optimization is not a trivial extension of existing results

Weakness:
- Presentation: the paper takes up too much space introducing the existing MatrixRL paper. This can probably be compressed.
- The reviewer expects more explanation about why the simultaneous model is studied, and why it is significant. It is not entirely clear to the reviewer in what situation the simultaneous acting model makes sense. If it is a distributed model, then a discussion about the communication model should be presented. In the introduction, the authors should probably give some in-depth discussion about this.
- A lack of lower bound: as discussed in the paper, the presented upper bound is only superior in certain parameter regimes. But there is a lack of hardness results to justify the significance of the current result.

* A question: for the "S" factor in the confidence bound, it does not mean the number of states, right? If not, it might be good to change it to another symbol as "S" would introduce confusion.

---

### Review · Reviewer_MxEo · 2022-10-10

**Summary Of Contributions:**

1. an improved algorithm analysis in Yang & Wang (2019) via a lazy version of the determinant lemma.
2. A new model for multi-task RL assumes the matrix in Yang & Wang (2019) admits a low-rank structure and all tasks share a common projection operator.
3. A new algorithm exploits the low-rank structure and is shown to enjoy a better regret than treating each task independently.
4. An computationally efficient method for the case when the state-action space is finite.


**Requested Changes:**

1. I found the method in Section 5 unsatisfying because it requires a finite state-action space. I think requiring a finite action space is still fine, but a finite state is far from making this method practical. Adding some experiments (even on some simulation environments) will strengthen this paper. Although the optimization problem can be intractable in the worst case, the authors can still resort to some heuristic solvers to show in practice, and the algorithm can still give reasonable performance.

2. Adding a lower bound to show the regret upper bound is (nearly) tight.

**Strengths And Weaknesses:**

Strengths:
1. A new formulation of multi-task RL.
2. A new algorithm is proposed, and the regret bound looks tight.
3. The overall writing is clear.

Weaknesses:
1. The proposed methods are not computationally efficient.
2. No experiments.
3. No lower bound.

---

### Review · Reviewer_HUgL · 2022-10-13

**Summary Of Contributions:**

This paper presents a theoretical result on the benefit of multi-task reinforcement learning in Yang and Wang's linear MDP setting, where it is further assumed that the matrix M has a low-rank factorization in which the projection operator is shared across all tasks. The theoretical rate is solid and achieves improvement over learning the tasks independently.

**Requested Changes:**

As discussed above, I encourage the authors to add a matching lower bound to the paper, but I would recommend for acceptance regardless.

**Strengths And Weaknesses:**

Strengths: the presentation is clean, and the theoretical results are solid. I particularly appreciate the improvement in computational efficiency in section 5, which is a good step toward making the algorithm applicable to real applications.

Weakness: Currently, the lower bound is missing. For a journal publication, a complete theoretical picture with matching upper and lower bounds would be much more appreciated than leaving it for future research. Especially for this problem, since a lower bound shouldn't be too hard to construct, as already hinted by the authors, I would encourage the authors to make the additional effort to make it a good paper.

---

### Decision · Action_Editors · 2022-12-20

**Recommendation:** Reject

**Comment:**

While this paper has some good results (the removal of a $\sqrt{H}$ factor, and improved regret versus independently learning tasks), I do not believe it should be accepted without a major revision. The reviewers are divided, with one weakly supporting acceptance and the other two weakly supporting rejection. When reading the below, I would like the authors to keep in mind that they did not respond to the reviews. Consequently, I am unclear on their perspective regarding the various changes requested by reviewers, and this no doubt has an effect on the decision made on this work.

First, all three reviewers mentioned wanting a lower bound (one asks for a matching one, one asks to show some lower bound, and another asks for at least a discussion of a lower bound). The authors should at least discuss lower bounds, even if not proving one.

Following the advice of Reviewer aEHw would make this work much stronger. In particular, for a paper of this type, a technical overview in the Introduction is essential. Also, it is important to motivate the simultaneous multi-task model (i.e., motivating simultaneity vs sequentiality). I feel that it is more common to consider sequentially learning the multiple tasks (one task after the other). Finally, it would be very helpful to give a discussion of your bound immediately after Theorem 4.1.

In addition, Reviewer MxEo mentions that the proposed algorithms are not computationally efficient. Based on my own read of the paper, I wondered why it is reasonable to assume access to a bilinear optimization oracle. There should be more discussion of this point, including (if they exist) references for why access to such an oracle is reasonable in certain settings of interest; that is to say, are there situations in which such an oracle can be efficiently implemented?

Finally, my own advice is to please try to improve the presentation. At times, you give results that may be quite important, but there is little to no discussion as to their wider significance. For instance, you mention that Lemma 4.3 is one of your main results. Yet, this result is immediately followed by a proof, with no further discussion of the result. Also, I wonder if Theorem 4.7 and Corollary 4.8 can be given earlier, rather than waiting until the technical supporting lemmas have been presented. With an expanded Introduction that includes the aforementioned technical overview, you could also give informal versions of the results and some discussion of them in the intro itself.

Also, some easier-to-address things:
- Just before Section 2, $H$ is used without introduction.
- At the bottom of page 2, $d'$, $d$, and $r$ are used without introduction.
- I find the last sentence in the penultimate paragraph of Section 2 to be rather odd: "… should be thought of as a successor works to ours." It seems strange that existing works would be considered successor works. If this is true, some more explanation would really help.

**Audience:**

From the Reinforcement Learning Theory side, I believe there would be a sizeable audience for this work. From the multi-task learning side, I am less sure, due to limited motivation for the simultaneous multi-task setting.

**Claims And Evidence:**

The claimed regret bounds are well supported by the proofs. One reviewer has concerns about the practical performance of the method; however, as far as I am aware, this does not contest the claims of the paper (as the authors never claim to have a computationally efficient method unless there is access to a bilinear optimization oracle).